# Systematic mapping of BCL-2 gene dependencies in cancer reveals molecular determinants of BH3 mimetic sensitivity

Ryan S. Soderquist[1], Lorin Crawford [2,3], Esther Liu[1], Min Lu[1], Anika Agarwal[1], Gray R. Anderson[1], Kevin H. Lin[1], Peter S. Winter[1], Merve Cakir[1] & Kris C. Wood[1]

While inhibitors of BCL-2 family proteins (BH3 mimetics) have shown promise as anti-cancer agents, the various dependencies or co-dependencies of diverse cancers on BCL-2 genes remain poorly understood. Here we develop a drug screening approach to define the sensitivity of cancer cells from ten tissue types to all possible combinations of selective BCL-2, BCL-X$_L$, and MCL-1 inhibitors and discover that most cell lines depend on at least one combination for survival. We demonstrate that expression levels of BCL-2 genes predict single mimetic sensitivity, whereas EMT status predicts synergistic dependence on BCL-X$_L$+MCL-1. Lastly, we use a CRISPR/Cas9 screen to discover that BFL-1 and BCL-w promote resistance to all tested combinations of BCL-2, BCL-X$_L$, and MCL-1 inhibitors. Together, these results provide a roadmap for rationally targeting BCL-2 family dependencies in diverse human cancers and motivate the development of selective BFL-1 and BCL-w inhibitors to overcome intrinsic resistance to BH3 mimetics.

[1] Department of Pharmacology and Cancer Biology, Duke University, Durham, NC 27710, USA. [2] Department of Statistics, Duke University, Durham, NC 27710, USA. [3] Present address: Department of Biostatistics, Brown University School of Public Health, Providence, RI 02903, USA. Correspondence and requests for materials should be addressed to K.C.W. (email: kris.wood@duke.edu)

The process of intrinsic apoptosis is tightly regulated by the BCL-2 family of proteins. In human cancers, the anti-apoptotic BCL-2 proteins play a critical role in protecting cells, which are often "primed" for apoptosis, from committing to irreversible cell death[1]. To date, the most well described of the anti-apoptotic BCL-2 genes are BCL-2, BCL-$X_L$, and MCL-1, and recently, following over a decade of extensive research effort, potent and selective inhibitors of each of these proteins were developed. Much is known about the cancer types that respond well to selective BCL-2 inhibitors, and indeed the BCL-2 inhibitor venetoclax (ABT-199) is now FDA approved to treat certain leukemias such as chronic lymphocytic leukemia (CLL)[2,3]. In contrast, outside of a small number of studies in select cancer types, little is known regarding which cancers might respond well to single agent BCL-$X_L$ or MCL-1 inhibition[4–7]. Finally, to the best of our knowledge, no studies have systematically examined the dependencies of cancers on combinations of BCL-2 family proteins.

With these limitations in mind, we set out to address the following questions: What are the dependencies of diverse human cancers with respect to BCL-2, BCL-$X_L$, MCL-1, and their combinations? What are the molecular features of tumors that drive these dependencies? Finally, which cancers fail to respond to BH3 mimetics, and how can this intrinsic resistance be overcome? To answer these questions, we developed a screening strategy to assess the sensitivity of cancer cell lines to all possible combinations of a selective BCL-2 inhibitor (ABT-199), a selective BCL-$X_L$ inhibitor (WEHI-539), and a selective MCL-1 inhibitor (A-1210477). Using this approach, we mapped cellular dependencies and co-dependencies on BCL-2, BCL-$X_L$, and MCL-1 across a large number of primary and established cancer cell lines representing 10 distinct cancer types. These data provide new insights into the landscape of sensitivity to BH3 mimetics in human cancers, revealing molecular determinants of sensitivity and a role for a novel endoplasmic reticulum (ER) stress-epithelial-mesenchymal transition (EMT) axis in dictating the frequently observed synergy between BCL-$X_L$ and MCL-1 inhibitors in solid tumors. Collectively, these findings may help guide the use of BH3 mimetics as precision therapies in defined cancers.

## Results

**Mapping of BCL-2 gene dependencies.** To begin, we first made several assumptions regarding the BH3 mimetic drugs ABT-199, WEHI-539, and A-1210477 based on prior literature and our own experience. First, we elected to perform screens using a concentration of 1 μM for both ABT-199 and WEHI-539, as complete target inhibition is observed at these concentrations, and concentrations above this level may have off-target effects or may not be achievable in patients. A-1210477 is a first-in-class probe compound, and as such is less potent than ABT-199 or WEHI-539. Therefore, a concentration of 10 μM was selected for this compound, as at this dose MCL-1 is fully inhibited without inhibitory effects on BCL-2 and BCL-$X_L$[8]. A drug panel consisting of all possible single, double, and triple agent combinations of these drugs, at these concentrations, was then constructed and assayed in cell lines after a 72 h treatment using a conventional viability assay (see Methods) (Fig. 1a). To ensure that this assay accurately reveals BCL-2 family dependencies, we assembled several cell lines previously reported to be dependent on BCL-2, BCL-$X_L$, MCL-1, or combinations of these proteins, then verified the recovery of expected dependencies (Fig. 1b) [6,9–11]. In prior studies, we identified Panc 03.27 cells as BCL-$X_L$ dependent, and as such this line was included as a control. To further validate this BCL-2 family dependency assay, we compared its results to conventional BH3 profiling assays (Supplementary Fig. 1A–C).

Consistent with the reported selective, on-target activities of the BH3 mimetics above, these assays revealed that BCL-$X_L$ dependency levels from viability assays correlate strongly on a cell line by cell line basis with the activity of the HRK peptide, which selectively inhibits BCL-$X_L$. Similarly, MCL-1 dependency correlated with the activity of the NOXA peptide, a selective and direct inhibitor of MCL-1[1]. Lastly, we tested structurally independent BCL-$X_L$ (A-1331852) and MCL-1 (S63845) inhibitors, as well as a dual BCL-2/BCL-$X_L$ inhibitor (ABT-737), in cell lines exhibiting single agent BCL-2, BCL-$X_L$, or MCL-1 inhibitor sensitivity, or a cell line resistant to the inhibition of all three genes (Supplementary Fig. 2). Importantly, the sensitivity of each of these cell lines to these BH3 mimetics recapitulated the sensitivities observed following treatment with ABT-199, WEHI-539, or A-1210477, providing confidence that the latter drugs can be used as reliable probes of BCL-2 family dependencies[9–11].

Next, we used this assay to profile a large panel of 78 established cell lines from diverse cancer types, including acute myeloid leukemia (LAML), high-grade serous ovarian cancer (OV), colorectal adenocarcinoma (COAD), pancreatic ductal adenocarcinoma (PAAD), non-small cell lung cancer (LUAD), cutaneous melanoma (SKCM), liver hepatocellular carcinoma (LIHC), bladder cancer (BLCA), breast cancer (BRCA), and glioblastoma multiforme (GBM) (The Cancer Genome Atlas (TCGA) abbreviations used throughout; Fig. 1c and Supplementary Table 1). Further, we complemented these findings with assays performed in a panel of 13 primary patient-derived cultures (Fig. 1d), including primary COAD cells established following short-term in vivo propagation as patient-derived xenografts (PDXs) and primary PAAD cultures established directly from patient tumors[12,13]. In both cases, data from primary cultures closely resembled those from established cell lines, suggesting that apoptotic regulatory mechanisms are largely maintained during culture. Further, the intermediate sensitivity of JH4.3 cells to BCL-$X_L$ inhibition, initially observed in vitro, was recapitulated in an in vivo xenograft model, suggesting correspondence between in vitro and in vivo sensitivities in these models (Fig. 1e).

Collectively, pan-cancer cell line response behaviors revealed a number of notable patterns. First, whereas many of the LAML cell lines tested exhibited some dependency on BCL-2, a finding that coheres with published literature[14], the majority of solid cancers did not. Exceptions to this observation were SKCM cell lines and one LIHC cell line (SNU 423), which exhibited a modest BCL-2 dependency. This finding likely explains why single agent BCL-2 inhibitors such as ABT-199 have been less efficacious in solid cancers compared to LAML. LAML cells did not depend on BCL-$X_L$ for survival but did exhibit a robust MCL-1 dependency, again consistent with recent evidence[15]. In solid cancer cell lines, a fraction were dependent on BCL-$X_L$, MCL-1, or the combination of the two. In particular, we observed a specific dependence on BCL-$X_L$ in select tumor types, in particular in subsets of BLCA, SKCM, and PAAD lines. This finding was unexpected given that solid tumors have only rarely been associated with single agent sensitivity to BH3 mimetics and suggests the possibility of defining subsets of these tumors vulnerable to selective BCL-$X_L$ inhibition. We also observed MCL-1 dependence in subsets of BRCA, SKCM, and PAAD cell lines, with the sensitivity of BRCA lines to single agent MCL-1 inhibition being consistent with recent reports[16]. Finally, and surprisingly, we observed that 49 of 78 cell lines were sensitive (defined as a viability loss of >50%) to the combined inhibition of BCL-$X_L$ + MCL-1. Consistent with the lack of activity of single agent BCL-2 inhibition, the addition of ABT-199 to the three conditions above (i.e. BCL-2+BCL-$X_L$, BCL-2+MCL-1, and the triple combination) did not cause a significant increase in viability loss, suggesting that BCL-2 does not significantly contribute to intrinsic

BH3 mimetic resistance in solid cancers, and that BCL-2 selective inhibitors may be unlikely to improve the apoptotic responses of solid tumors to various drug therapies. Finally, we observed that cell lines from OV, COAD, and LUAD tumors were frequently insensitive to all combinations of BH3 mimetics, suggesting either lower overall degrees of apoptotic priming or different anti-apoptotic BCL-2 protein dependencies in these tumors. To distinguish between these two possibilities, we selected cell lines that were singly dependent on BCL-2, BCL-$X_L$, or MCL-1 (single-gene), synergistically dependent on BCL-$X_L$ + MCL-1, or resistant to all tested BH3 mimetics. BH3 profiling was then performed on these cell lines using the BIM and PUMA peptides, which measure a cell's ability to undergo apoptosis and overall BCL-2 priming, respectively (Supplementary Figure 3). Importantly, resistant cell

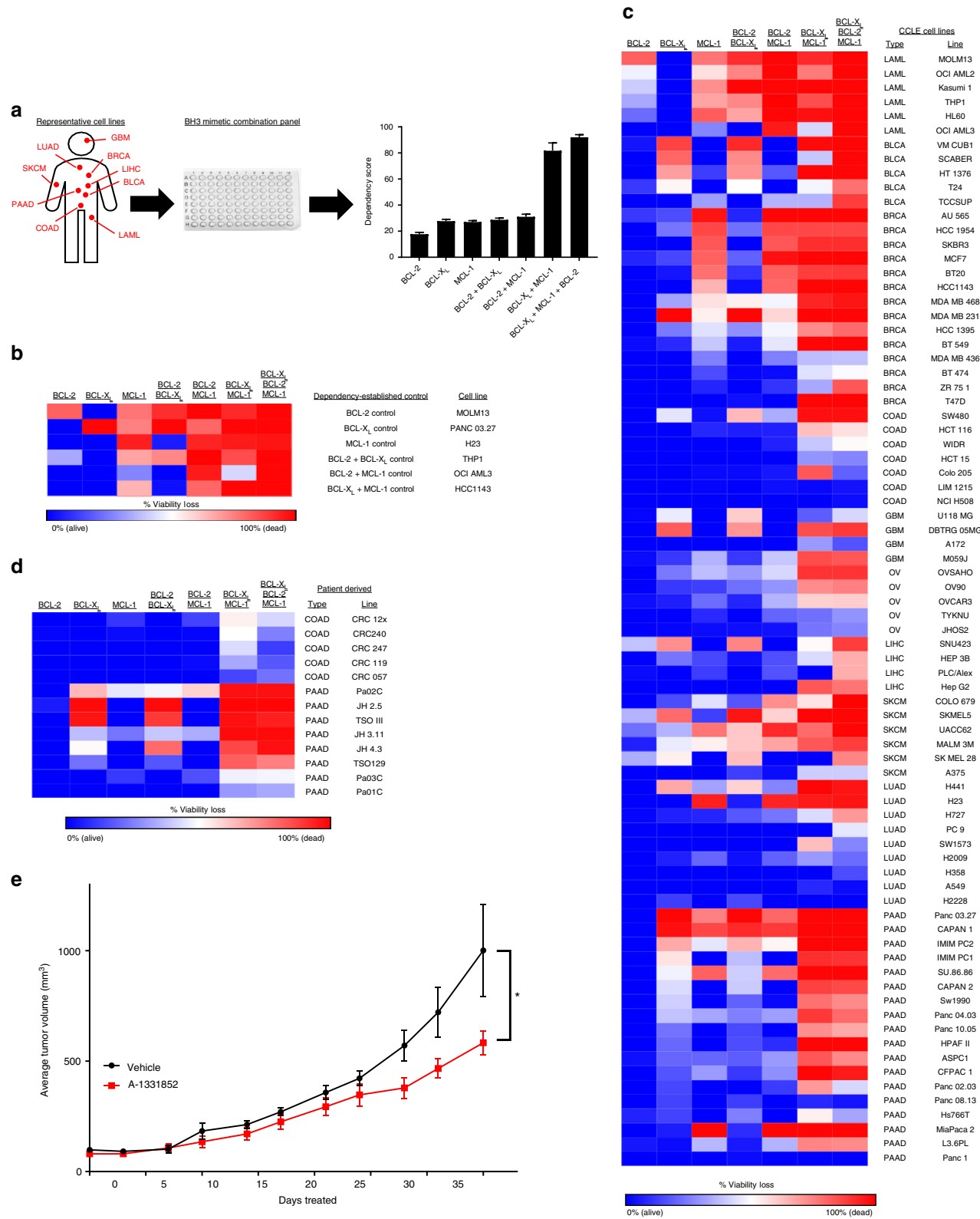

lines were still depolarized by the BIM peptide, indicating that they have intact BAX/BAK machinery and are capable of undergoing apoptosis. Similarly, the PUMA peptide also induced depolarization in the resistant cell lines, indicating that these cell lines indeed have some degree of dependence on BCL-2 family anti-apoptotic proteins. However, the single-gene dependent group of cell lines exhibited a significantly higher PUMA depolarization signal than either the synergistic or the resistant groups, indicating that this group of cell lines had an abnormally high amount of BCL-2 family priming. Taken together, these data indicate that resistant cell lines (i.e., cell lines that do not respond to any of the tested combinations of BH3 mimetics) are apoptotically competent and exhibit comparable overall BCL-2 family priming relative to the cell lines that synergistically respond to BCL-$X_L$+MCL-1 co-inhibition. As such, these resistant cell lines likely rely on some combination of additional BCL-2 genes (e.g., BFL-1, BCL-w, etc.) which are not inhibited by the available BH3 mimetics. In summary, the results of this large-scale profiling effort reveal that many solid tumor types are dependent upon some combination of BCL-2 family anti-apoptotic proteins, including a large subset of tumors with targetable dependencies on BCL-$X_L$, MCL-1, or the combination, motivating the search for specific biomarkers of these vulnerabilities.

**Molecular determinants of single-gene dependencies**. Next, we sought to establish the molecular determinants of the key responses observed in our cell line profiles: single agent BCL-$X_L$ and MCL-1 dependency and synergy between BCL-$X_L$ and MCL-1. Two of the most dominant phenotypic drivers in a given cancer are oncogenic mutations (oncogene or tumor suppressor status) and tissue of origin. As such, we first sought to determine which of these two properties best predicted BCL-2 gene dependencies. To address this question, we filtered our list of 78 cell lines to only those that are in the Cancer Cell Line Encyclopedia (CCLE) and therefore can be linked to publicly available genomic annotation data. These cell lines were then grouped based on their pre-dominant BCL-2 gene dependencies (with a 25% viability loss threshold): BCL-$X_L$ dependent, MCL-1 dependent, BCL-$X_L$+MCL-1 co-dependent, or mostly resistant (Fig. 2a). We selected 10 of the most commonly mutated genes in human cancers (TP53, PIK3CA, PTEN, KRAS, EGFR, NF1, BRAF, RB1, ATM, BRCA2, and BRCA1) and determined the status of each of these genes in all cell lines. To determine if tissue of origin or mutation status was a better predictor of dependency phenotype, we first performed a linear regression analysis (Supplementary Table 2) for each dependency phenotype (BCL-$X_L$ dependence, MCL-1 dependence, etc.), and the goodness-of-fit for each model (Fig. 2b) was used to determine the better predictor of response. In all cases, tissue of origin was a superior predictor relative to oncogene/tumor suppressor mutation status, as indicated by higher $R^2$ values.

A key, distinguishing feature of cells from distinct tissues are their unique gene expression patterns. We and others have previously demonstrated that expression levels of BCL-2 genes often correlate, and at times even drive, dependencies on various BCL-2 genes. For example, a recent report demonstrated that expression of NOXA elicited a dependence on BCL-2 in a subset of neuroblastomas[17]. We therefore hypothesized that tissue of origin may dictate BCL-2 family dependencies via its effects on BCL-2 family gene expression patterns. To determine if expression levels of BCL-2 genes predicted single-gene dependencies across diverse cell lines, we mined mRNA expression levels of the predominate BCL-2 family genes for each cell line, then performed a linear regression analysis to identify those strongly correlated with functional dependencies (Fig. 2c and Supplementary Table 3). In agreement with prior reports, we found that BCL-2 gene expression positively and strongly correlated with BCL-2 dependency across cell lines. MCL-1 dependence is strongly associated (anti-correlated) with BCL-$X_L$ mRNA levels, a finding consistent with recent data from our group and others[4–6]. Lastly, expression levels of NOXA strongly correlated with BCL-$X_L$ dependence, with BCL-$X_L$ dependent cell lines having higher expression levels of NOXA than non-dependent lines. To confirm the functional relevance of the identified associations relevant to BCL-$X_L$ and MCL-1 dependencies, we first stably transduced three cell lines with a strong MCL-1 dependence (Mia Paca-2, H23, and SK-BR-3) using a BCL-$X_L$ overexpression construct, then treated cells with A-1210477 or the combination of this drug with WEHI-539 (Fig. 2d). Consistent with the correlation data, overexpression of BCL-$X_L$ reduced sensitivity to the MCL-1 inhibitor, resistance that could be reversed with the addition of the BCL-$X_L$ inhibitor. Similarly, three BCL-$X_L$ dependent cell lines (HT-1376, SCaBER, and T24) were stably transduced with shRNA targeting NOXA. Knockdown of NOXA expression attenuated sensitivity to BCL-$X_L$ inhibition, protection that could be reversed with the addition of A-1210477.

These findings demonstrate that across heterogeneous solid tumors from diverse tissue and oncogenic mutational back-grounds, sensitivity to single agent BCL-$X_L$ and MCL-1 inhibitors is strongly associated with the expression levels of defined BCL-2 family members. To determine whether the expression levels of these genes have predictive value, we used a sliding scale analysis to identify threshold expression levels of BCL-$X_L$ and NOXA that best segregated tumor cell lines on the basis of MCL-1 or BCL-$X_L$ dependency, respectively. This analysis identified an expression threshold for NOXA that robustly segregated WEHI-539 responsive and non-responsive lines (Fig. 2e, $p = 0.005$). Further, this analysis also identified a BCL-$X_L$ expression threshold that robustly segregated A-1210477 responsive and non-responsive lines (Fig. 2e, $p < 0.0001$). To independently validate this finding, we used data from a recent, large cell line chemogenomic profiling effort which determined the sensitivity of hundreds of cell lines to the dual BCL-2 and BCL-$X_L$ inhibitor ABT-263, among other agents[18]. The NOXA expression threshold identified

**Fig. 1** Systematic mapping of BCL-2 gene dependencies. **a** Workflow. Cell lines from 10 cancer types were treated with all combinations of the BCL-$X_L$ inhibitor WEHI-539 (1 μM), the BCL-2 inhibitor ABT-199 (1 μM), and/or the MCL-1 inhibitor A-1210477 (10 μM) for 3 days in a 96 well plate and assessed for changes in viability via Cell Titer-Glo (CTG). **b** A heatmap showing BCL-2 gene dependencies from six control cell lines that have known dependencies on: BCL-2 (MOLM13), BCL-$X_L$ (Panc 03.27), MCL-1 (H23), BCL-2 + BCL-$X_L$ (THP1), BCL-2 + MCL-1 (AML3), and BCL-$X_L$ + MCL-1 (HCC1143). Percentage viability loss is calculated as 100 – (the % viability signal determined from the CTG assay) and is colored from 0 % viability loss (blue) to 100 % viability loss (red). **c** Heatmap of BCL-2 gene dependencies in cell lines representing: acute myeloid leukemia (LAML), high-grade serous ovarian carcinoma (OV), colorectal adenocarcinoma (COAD), pancreatic ductal adenocarcinoma (PAAD), non-small cell lung carcinoma (LUAD), melanoma (SKCM), liver (LIHC), bladder (BLCA), breast (BRCA), glioblastoma (GBM). **d** Patient-derived cell lines from COAD or PAAD tumors were assayed for BCL-2 gene dependencies. **e** An in vivo xenograft model of JH4.3 cells grown in athymic mice. Once tumors reached 100 mm³, mice (6-7 per group) were treated with vehicle or the BCL-$X_L$ inhibitor A-1331853 (25 mg/kg, qd) until the vehicle group reached 1000 mm³ (35 days). A 2-way ANOVA between vehicle ($n = 5$) and treated ($n = 7$) groups yielded a significant $p$-value (*, $p = 0.005$)

above also effectively segregated sensitive and resistant lines ($p <$ 0.0001), with lines expressing high levels of NOXA being more sensitive to ABT-263 than cell lines below the threshold, and sensitive lines exhibiting submicromolar median inhibitory concentration-50% (IC50) values (Fig. 2f). Together, these data demonstrate that expression levels of specific BCL-2 family members can be used to identify solid tumors sensitive and resistant to single agent BH3 mimetics, independent of tissue type or oncogenic mutational background.

**EMT underlies synergistic co-dependence on BCL-X$_L$+MCL-1.** Perhaps the most striking and unexpected finding from our cell line profiling effort (Fig. 1c) is the extent to which inhibiting BCL-X$_L$, MCL-1, or the combination is efficacious in solid

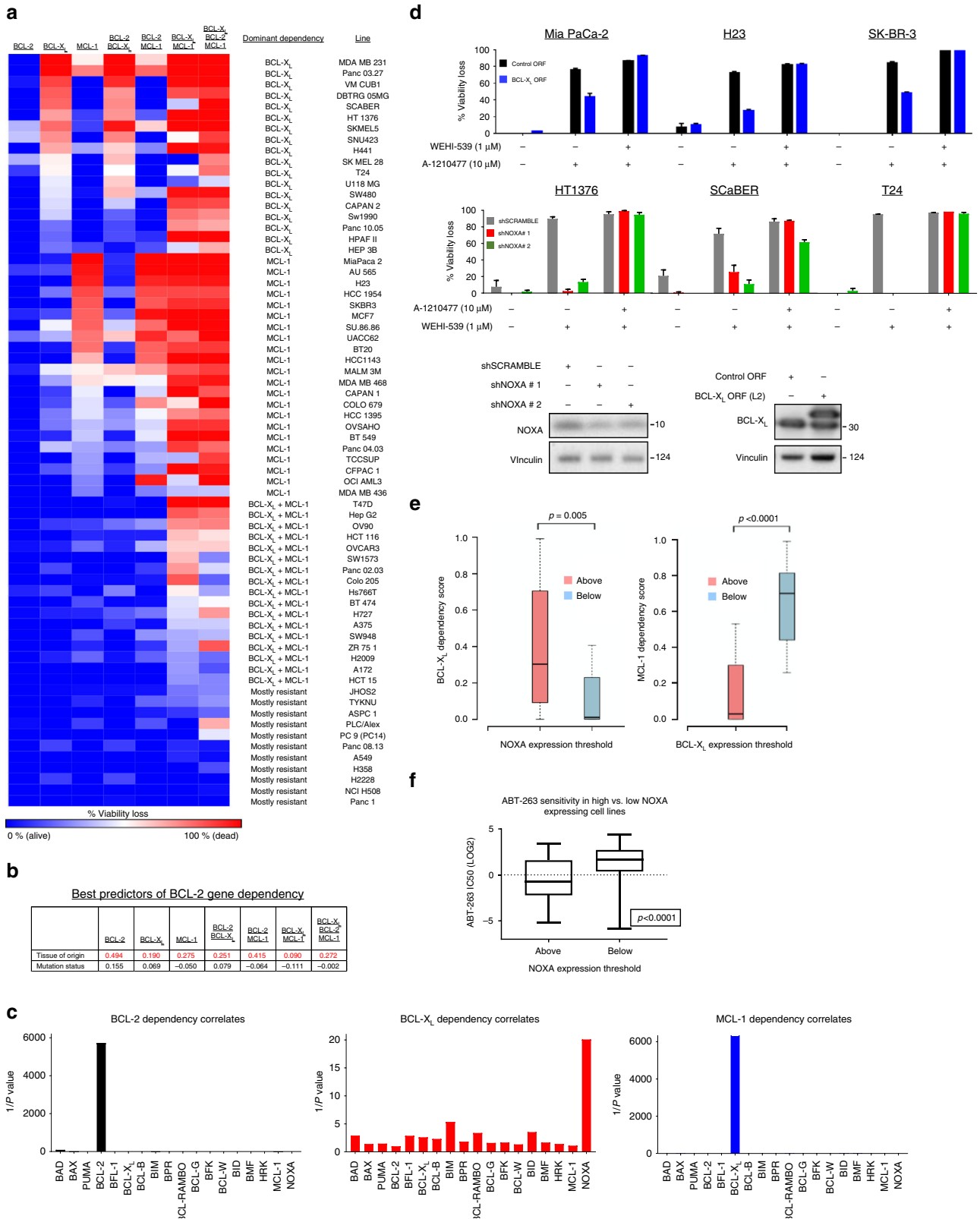

tumor-derived lines. Specifically, of the established cell lines for which genomic data are available, only 7 of 69 lines failed to respond (<20% viability loss) to some combination of these inhibitors. Approximately 50% of these responding lines exhibited single agent sensitivity to BCL-$X_L$ or MCL-1 inhibitors, while the remaining lines showed pronounced synergy between BCL-$X_L$ and MCL-1 inhibitors, with the combination being substantially more potent than expected on the basis of additivity alone (Fig. 3a). To identify signaling events that may underlie this BCL-$X_L$/MCL-1 synergy, gene set enrichment analysis (GSEA) was performed (Fig. 3b). Surprisingly, this analysis revealed that the top four pathways associated with BCL-$X_L$/MCL-1 inhibitor synergy were all related to the epithelial–mesenchymal transition (EMT). To further examine this relationship, we formulated "mesenchymal" and "epithelial" expression signatures (see Methods). Using these signatures, we observed that BCL-$X_L$/MCL-1 inhibitor synergy is anti-correlated with the mesenchymal state and positively correlated with the epithelial state, both across individual cell lines (Fig. 3c) and in tissue-based groupings of cell lines (Fig. 3d).

To functionally confirm these findings, we targeted E-cadherin with lentiviral shRNAs to induce EMT via the release of β-catenin[19]. In five cell lines from four tissues of origin displaying variable levels of BCL-$X_L$/MCL-1 inhibitor synergy, EMT activation via E-cadherin knockdown significantly reduced synergy (Fig. 3e, f). Taken together, these data suggest that epithelial cells maintain a state wherein apoptosis is buffered by both BCL-$X_L$ and MCL-1, and inhibition of both molecules is required to sufficiently induce apoptosis. Upon transition to a mesenchymal state, this dual dependence is lost. Due to the robustness of this phenotypic switch, we next sought to understand the molecular mechanism(s) that might explain this finding.

**EMT increases dependence on BCL-$X_L$ through PERK signaling.** Triggering an EMT could cause a decrease in the synergy associated with the BCL-$X_{L +}$ MCL-1 inhibitor combination by either causing a decrease in sensitivity to the combination or by increasing sensitivity to the BCL-$X_L$ or MCL-1 inhibitor alone. In the cell line panel above, forcing EMT via shE-cadherin transduction failed to either decrease sensitivity to the drug combination or increase sensitivity the MCL-1 inhibitor (Supplementary Fig. 4). However, an increase in sensitivity to the BCL-$X_L$ inhibitor was observed in all tested lines (Fig. 4a). Consistent with this finding, BCL-$X_L$ dependence was significantly and positively correlated with the mesenchymal signature across individual cell lines ($p = 0.0009$) and in tissue-based groupings of cell lines (Fig. 4b). To further test this correlation between mesenchymal status and BCL-$X_L$ dependence, we selected a

canonical EMT gene, Slug (SNAI2), and probed for its expression via western blotting in 15 representative cell lines (Supplementary Fig. 5). Slug protein expression was correlated with BCL-$X_L$ dependence (Fig. 4c). Collectively, these data support a model whereby EMT causes a loss of BCL-$X_{L +}$ MCL-1 inhibitor synergy via an increase in dependency on BCL-$X_L$ alone. Thus, EMT status, as measured by a mesenchymal signature or Slug expression, predicts a cell's position along the continuum between BCL-$X_L$ dependence and BCL-$X_{L +}$ MCL-1 synergistic co-dependence.

The regulation of apoptosis by BCL-2 family proteins is a tightly regulated process which can be altered dramatically by stochiometric changes in these proteins' expression levels or cellular availability. Therefore, the simplest explanation for how EMT status could shift BCL-$X_L$ dependence would be that this process somehow impacts the expression/availability of a BCL-2 protein(s). Since BCL-$X_L$ and NOXA expression were the strongest correlates of BCL-$X_L$/MCL-1 dependence, we first analyzed protein expression for these genes following induction of an EMT (via shE-cadherin). In PC-9 cells, E-cadherin knockdown did not affect BCL-$X_L$ protein expression, but caused a dramatic increase in NOXA protein expression (Fig. 5a). This finding is in line with our previous finding that knockdown of NOXA is sufficient to protect BCL-$X_L$ dependent cell lines from the effects of BCL-$X_L$ inhibition (Fig. 2d). While there are several signaling pathways that are known to induce NOXA, we sought to identify which of these pathways may also be regulated by EMT. Toward this end, a recent publication demonstrated that EMT signaling can activate the PERK signaling pathway (a component of the ER stress pathway)[20]. Importantly, several reports have also demonstrated that PERK/ER stress signaling can increase the expression of NOXA via a pathway involving activation of PERK, phosphorylation of eiF2α (which inhibits cap-dependent mRNA translation), induction of ATF4 and ATF3, and ATF4/ATF3-dependent induction of NOXA expression[21–24]. Indeed, this pathway was induced in PC-9 cells transduced with multiple independent shE-cadherin hairpins, suggesting this might be the mechanism of NOXA induction (Fig. 5b). This hypothesis was confirmed by treating cells with the PERK inhibitor GSK2606414, which prevented the activation of the PERK signaling pathway and the induction of NOXA (Fig. 5c). Together, these data support a model in which epithelial cells depend synergistically on BCL-$X_L$ and MCL-1 for survival, whereas in mesenchymal cells the induction of the PERK signaling pathway, and consequently of NOXA, inhibits MCL-1, resulting in BCL-$X_L$ dependence (Fig. 5d).

**Resistance to BCL-$X_{L +}$MCL-1 Inhibition from BFL-1 or BCL-w.** Although most tested cell lines responded to some combination

**Fig. 2** Identification of specific BCL-2 genes as predictors of single BCL-2 gene dependencies. **a** Grouping of CCLE cell lines. CCLE cell lines (excluding BCL-2 dependent cell lines) with expression data were grouped based on BCL-2 gene dependencies. The lines were clustered broadly into four distinct groups: BCL-$X_L$ or MCL-1 dependent (>25% viability loss following BCL-$X_L$ or MCL-1 inhibition), BCL-$X_L$+MCL-1 co-dependent (>25% viability loss from the combination), and mostly resistant. **b** Goodness of fit analysis between BCL-2 gene dependency data and tissue of origin/mutation status. $R^2$ values from the linear regression analysis were plotted for each phenotype (red = higher $R^2$ value). **c** Correlations between BCL-2 gene expression and dependence on BCL-2, BCL-$X_L$, or MCL-1. 1/$p$-values from the linear regression analysis were plotted to visualize the best correlate for each phenotype. **d** MCL-1 dependent (Mia PaCa-2, H23, SK-BR-3) or BCL-$X_L$ dependent cell lines were transduced with V5-tagged BCL-$X_L$ overexpression ORF or two shRNAs targeting NOXA, respectively, and representative expression changes were quantified via western blot. Cells were treated with WEHI-539 (1 μM), A-1210477 (10 μM), or both and viability loss was analyzed via CTG assay ($n = 3$) after 3 days in drug; data are presented as mean viability loss ± SEM. **e** A sliding scale analysis was performed in the CCLE cell lines used in this study to identify the expression levels of NOXA or BCL-$X_L$ that best segregated cell lines as resistant or sensitive to BCL-$X_L$ or MCL-1 inhibition. BCL-$X_L$ or MCL-1 dependency scores are the percentage viability loss in cells treated with BCL-$X_L$ or MCL-1 inhibitors, respectively. A NOXA or BCL-$X_L$ mRNA expression threshold of 10.33 (LOG$_2$ score, TCGA expression database) or 6.44, respectively, yielded the most significant difference in BCL-$X_L$ ($p$-value = 0.005), or MCL-1 ($p$-value < 0.0001) dependence, respectively. Cell lines above the NOXA or below the BCL-$X_L$ expression thresholds demonstrated increased BCL-$X_L$ or MCL-1 dependence respectively. **f** The NOXA threshold value (10.33) was used to separate cell lines from an independent dataset. A t-test comparison of ABT-263 IC50 values (LOG$_2$) in cell lines below or above the NOXA threshold

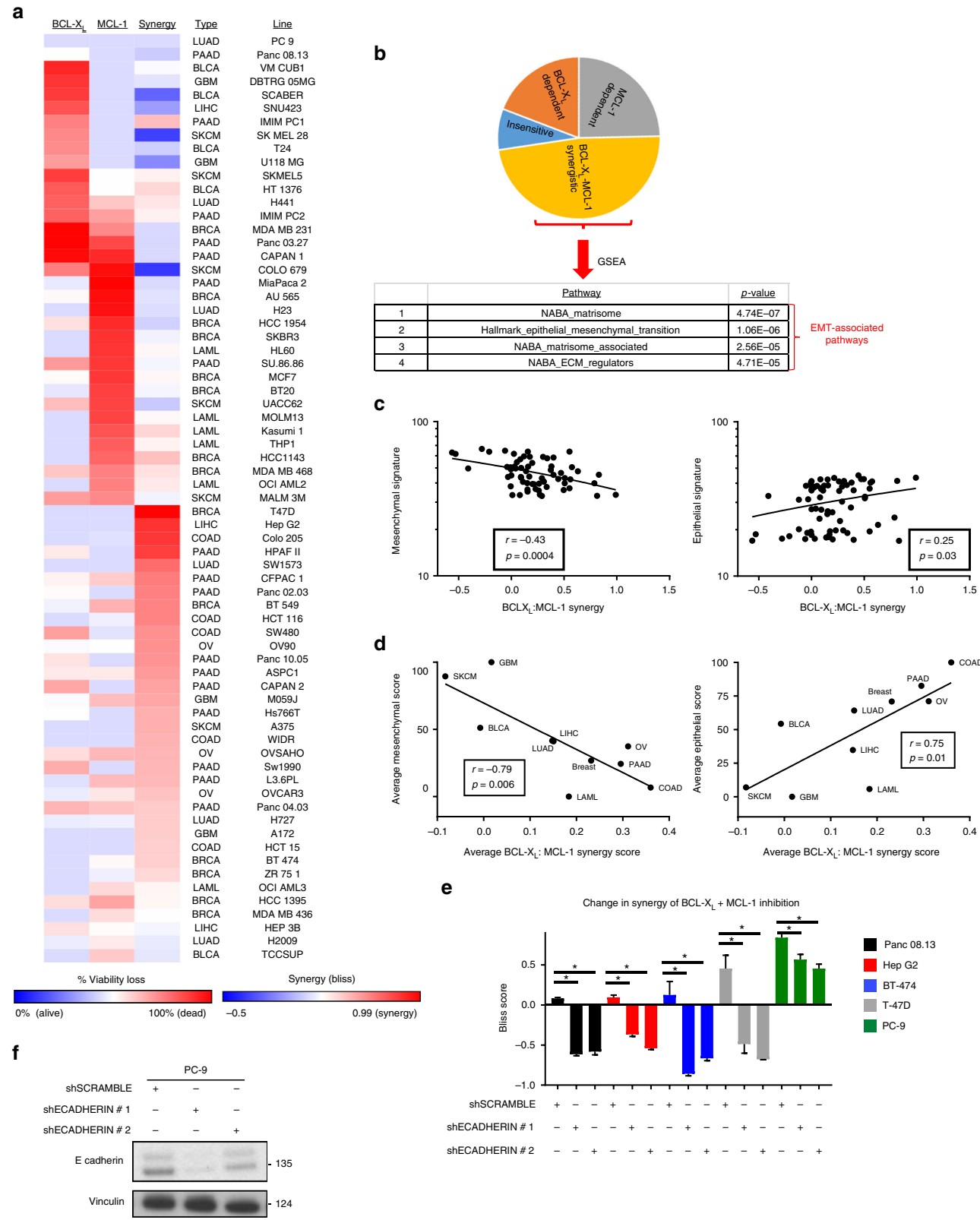

of BCL-2/BCL-X$_L$/MCL-1 inhibitors, we noted a minority of lines that were innately resistant (<10% viability loss) to every combination. This implies that there are additional mechanisms that protect these cell lines from apoptosis, even when BCL-2, BCL-X$_L$, and MCL-1 are inhibited. To find additional molecular targets to sensitize these resistant cell lines, we employed a negative-selection CRISPR screen using a library of sgRNAs that target a variety of cancer-related genes, including BCL-2 genes. The innately resistant cell line TYKNU was transduced with the CRISPR library at a low multiplicity of infection of 0.3, which

**Fig. 3** EMT status predicts BCL-X$_L$ + MCL-1 synergy. **a** A heatmap of synergy scores (−0.5 to 0.99) for the combined inhibition of BCL-X$_L$+MCL-1, compared to the BCL-X$_L$ or MCL-1 dependency scores (0–100% viability loss) from each line. Synergy was calculated according to the Bliss equation. **b** A pie chart showing the distribution of BCL-X$_L$/MCL-1 dependent, BCL-X$_L$:MCL-1 synergistically co-dependent, or insensitive cell lines. A GSEA was performed on cell lines ranked by the synergy score of the BCL-X$_L$+MCL-1 co-inhibition and the top four enriched pathways are shown. **c** Mesenchymal and epithelial gene signature scores were calculated from each cell line and compared to the BCL-X$_L$:MCL-1 synergy scores from each cell line in a Pearson correlation analysis. **d** Tissue-average mesenchymal/epithelial signature scores were calculated from each tissue type and compared to the tissue average BCL-X$_L$:MCL-1 synergy score via Pearson correlation analysis. **e** Genetic validation of EMT status and BCL-X$_L$:MCL-1 synergy. Three cell lines from the BCL-X$_L$:MCL-1 synergy group (T47D, Hep G2, BT-474) and two resistant cell lines (Panc 08.13, PC-9) were forced to undergo an EMT via the genetic knockdown of E-cadherin. The transduced cell lines were treated with WEHI-539, A-1210477 or both for 3 days, and percentage viability loss was assessed via CTG. Synergy scores for all conditions were calculated ($n = 3$) to determine what effect forcing an EMT has on BCL-X$_L$:MCL-1 synergy and data are presented as mean synergy scores ± SEM. A student's $t$-test was performed between each shECADHERIN vs the shSCRAMBLE ($* = p < 0.05$). **f** An example of E-cadherin knockdown is shown via western blotting

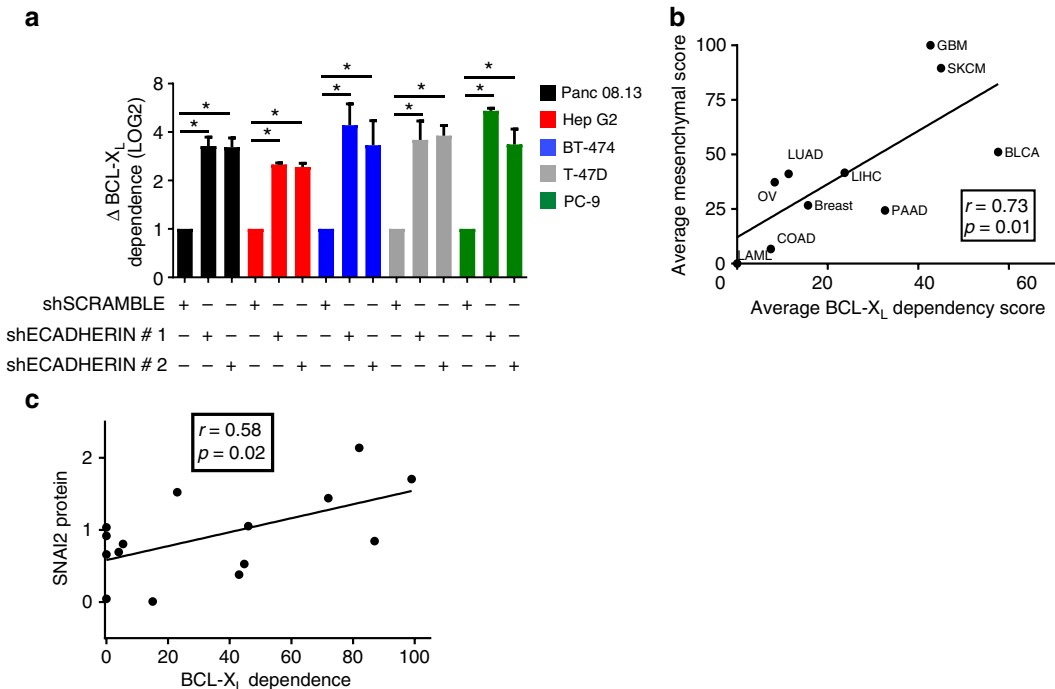

**Fig. 4** Mesenchymal status predicts BCL-X$_L$ dependence. **a**, Changes in BCL-X$_L$ dependence score in cells that underwent an EMT. The 5 cell lines that exhibited a decrease in BCL-X$_L$:MCL-1 synergy (Panc 08.13, Hep G2, BT-474, T-47D, PC-9) following EMT were treated with WEHI-539 for 3 days. Percentage viability loss was assessed via CTG and the change in BCL-X$_L$ dependence was calculated by normalizing the shECADHERIN transduced cells ($n = 3$) to the shSCRAMBLE ($n = 3$) and data are presented as the mean change (LOG2) in BCL-X$_L$ dependence ± SEM. A student's $t$-test was performed between each shECADHERIN vs shSCRAMBLE ($* = p < 0.05$). **b** The tissue-average mesenchymal and BCL-X$_L$ dependence scores were compared via Pearson correlation analysis. **c** The cell line protein expression (see Supplementary Fig. 5) of SNAI2 (SLUG), one of the genes from the mesenchymal signature, was correlated to BCL-X$_L$ dependence or BCL-X$_L$:MCL-1 synergy via a Pearson correlation analysis

reduces the likelihood of a single cell receiving multiple knock-outs. Screens were performed in the presence of combined BCL-X$_L$ and MCL-1 inhibition given that no additional toxicity was conferred by BCL-2 inhibition (Fig. 1). After a 2-week treatment with the drug combination, samples were sent for sequencing and the depletion scores for each gene were calculated relative to the vehicle control sample (Supplementary Table 4). Interestingly, among the top sensitizers were two relatively understudied BCL-2 genes: BFL-1, and BCL-w (Fig. 6a). The fact that these genes—which are not inhibited by any of the drugs used in the BH3 mimetic panel—scored as sensitizers suggests that they play important roles in preventing apoptosis in this cell line. To validate this finding, we used two individual sgRNAs for each gene to generate isogenic derivatives of TYKNU and two additional innately resistant cell lines (WiDR and A549), then subjected them to the BH3 mimetic panel (Fig. 6b, c). In TYKNU cells, knockout of either of these genes was sufficient to sensitize cells to varying degrees to inhibition of BCL-X$_L$, MCL-1, and the

combination, with BCL-w knockout having the most potent effects. Consistent with this finding, in both WiDR and A549 cells, BCL-w or BFL-1 knockout conferred sensitivity to BCL-X$_L$, MCL-1, and the combination, with BCL-w knockout conferring particularly strong sensitization (Fig. 6b, c). Together, these data demonstrate that intrinsic resistance to selective inhibition of BCL-X$_L$, MCL-1, and their combinations is mediated by the BCL-2 family members BFL-1 and BCL-w. Given that the knockout of these proteins led to complete viability loss in all intrinsically resistant cell lines following treatment with BH3 mimetics, these data provide a strong rationale for developing selective pharma-cological inhibitors of these relatively understudied BCL-2 family proteins.

Finally, throughout the course of this study, we observed several cell lines whose responses to certain BH3 mimetic combinations are exceptionally strong or exceptionally weak relative to the rest of the cell lines in that tissue type (Supplementary Table 5). Given that our results demonstrate

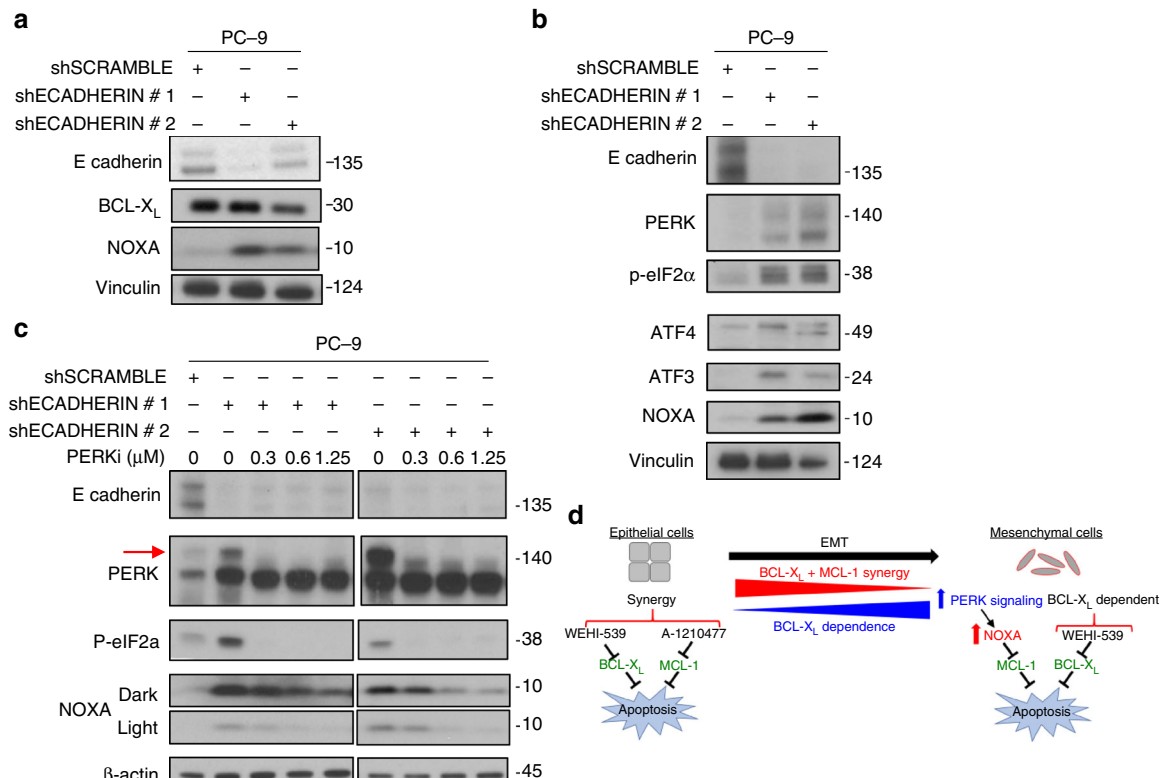

**Fig. 5** EMT induces NOXA expression and BCL-X$_L$ dependence via the PERK pathway. **a** Western blot data of PC-9 cells transduced with shECADERIN to induce an EMT. Expression levels of E-CADERIN, BCL-X$_L$, NOXA, and Vinculin are compared via western blotting in cells after a 3-day selection in puromycin. **b** Expression levels of PERK signaling markers in cells forced to undergo EMT. PC-9 cells were transduced with shECADERIN and after 3 days in puromycin, markers of the PERK signaling pathway (PERK, p-eIF2α, ATF4, ATF3, NOXA) were compared using western blotting. **c** Blocking PERK signaling prevents EMT-induced NOXA expression. PC-9 cells were transduced with shECADHERIN or control and immediately incubated in the indicated concentrations of the PERK inhibitor (PERKi) GSK2606414 for the remainder of the 3 days in puromycin. Loss of the PERK autophosphorylation band (red arrow) indicates that PERKi is effective at the concentrations used. **d** Model: Epithelial cells tend to exhibit a synergistic co-dependence on BCL-X$_L$+MCL-1, and therefore respond best to a combination of WEHI-539 and A-1210477. As cells become more mesenchymal like, either initially or via forcing an EMT, there is an increase in PERK signaling and PERK-dependent NOXA expression. This higher expression of NOXA renders cells more dependent on BCL-X$_L$ alone because the relatively higher amounts of NOXA neutralize MCL-1 in these cells. As such, the more mesenchymal cells are more responsive to WEHI-539 used as a single agent

that sensitivity to BH3 mimetic combinations is largely dependent upon the expression of BCL-2 family genes, we assessed whether these exceptionally responsive or non-responsive cell lines were outliers with regard to BCL-2 family expression (Fig. 6d). Specifically, for each outlier cell line, we calculated the percentage difference in both BCL-2 dependence and BCL-2 family gene expression compared to the averages for cell lines from its tissue of origin. In several cases, outlier expression of BCL-2 family genes was consistent with the observed changes in BCL-2 dependence. For example, OCI AML3 cells are resistant to BCL-2 inhibition (compared to the other tested LAML lines) and also express higher than tissue average levels of *MCL1* and *BFL1* mRNA. To functionally validate a selection of these exceptional outliers, we selected two cell lines: H441 and Panc 03.27. H441 cells are highly dependent on BCL-X$_L$ and also express higher levels of NOXA compared to tissue average (LUAD), a result which would be expected to lead to BCL-X$_L$ dependence via MCL-1 suppression. Similarly, PANC 03.27 cells are also exceptionally dependent on BCL-X$_L$ and express higher levels of NOXA compared to tissue average (PAAD). In each case, shRNA-mediated knockdown of NOXA was sufficient to reverse BCL-X$_L$ dependence, reverting these cells to a state more resembling the tissue average (Fig. 6e, f). Thus, the relationships between BCL-2 family expression and sensitivity the BH3 mimetics identified in this study can be used to identify exceptional responders, a finding that may allow for the

prospective identification of patients likely to respond to specific BH3 mimetic therapies. As a resource to guide the identification of additional outlier-expression relationships, we have broken down the heatmap of BCL-2 gene dependencies into separate "snapshot" figures based on tissue of origin, including mRNA expression data for key BCL-2 genes, in Supplementary Figures 6-15.

## Discussion

The identification of ABT-737 as the first true BH3 mimetic initiated a paradigm shift in terms of how we think about targeting apoptotic machinery in cancer cells. In the first decade following this shift, great progress was made targeting BCL-2 in a variety of leukemias, ranging from CLL to LAML. Since the development of ABT-737, much research has focused on synthesizing new BH3 mimetics which can inhibit other BCL-2 family members, with a particular emphasis on BCL-X$_L$ and MCL-1. Although many of these putative BH3 mimetics were later found to work through non-specific pathways[22–25], several on-target inhibitors of BCL-X$_L$ (WEHI-539, A-1155463, A-1331852) and MCL-1 (A-1210477, S63845) have recently been described[8,15,26,27]. Importantly A-1155463, A-1331852, and S63845 have all demonstrated safety and efficacy in early in vivo models, suggesting that these agents (or derivatives thereof) may someday make it to the clinic. These findings have culminated in the first FDA approved BCL-2 inhibitor (venetoclax), which has

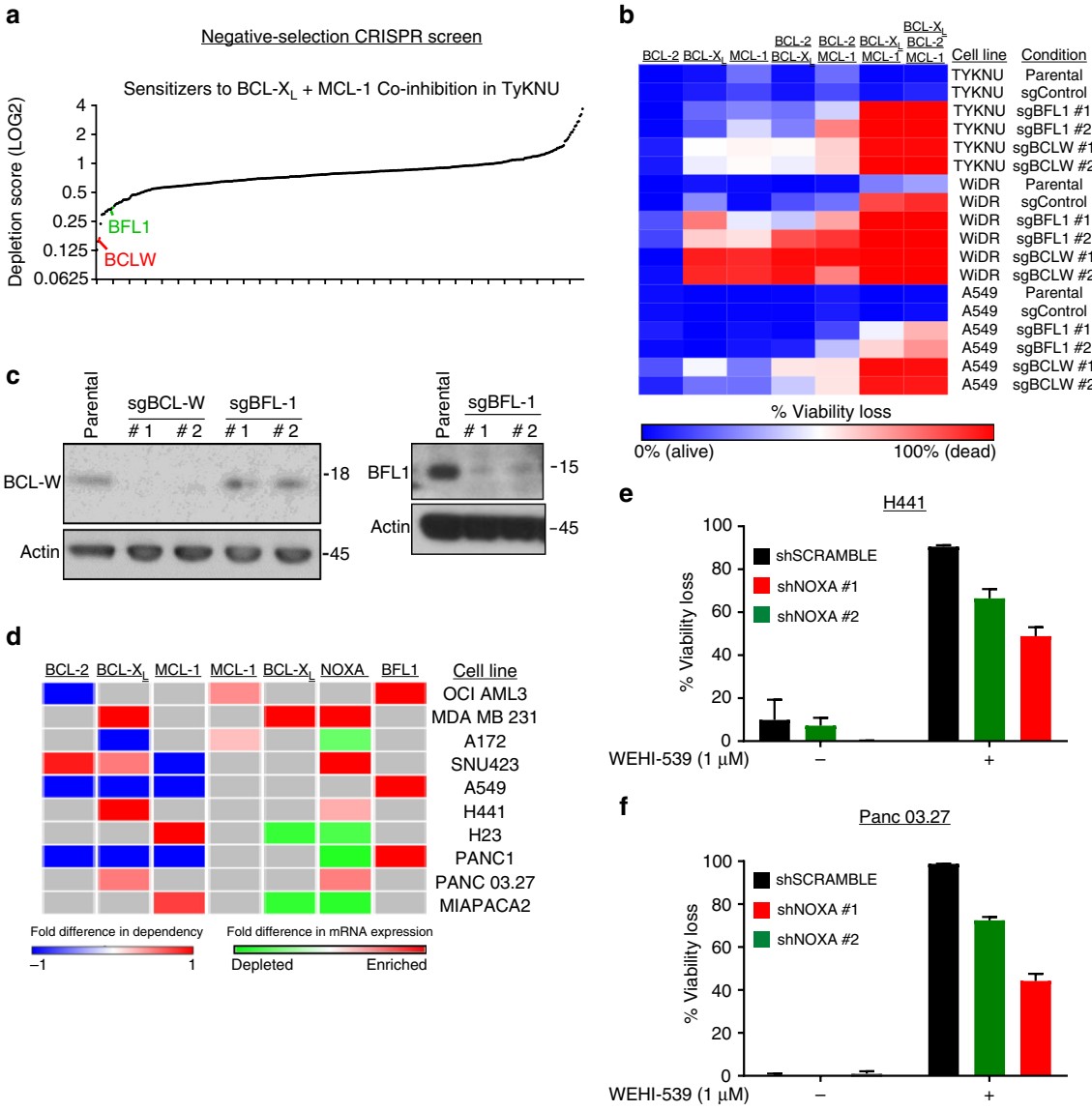

**Fig. 6** BCL-w and BFL-1 promote intrinsic resistance to BCL-X$_L$+MCL-1 inhibition. **a** A negative-selection CRISPR screen to identify sensitizers to combined BCL-X$_{L}$ + MCL-1 inhibition in a resistant cell line. The resistant cell line, TYKNU, was transduced with a CRISPR library containing cancer-relevant genes and grown in either vehicle (DMSO) or WEHI-539 (1 μM) + A-1210477 (10 μM) for 2 weeks. Genes whose depletion sensitized to the co-inhibition of BCL-X$_L$ + MCL-1 were deconvoluted via deep-sequencing of the barcoded CRISPR libraries from each sample. LOG$_2$ values of the depletion mean (DM) scores were plotted and relevant apoptotic genes were highlighted: red, BCL-w and green, BFL-1. **b** Validation of hits from screen. Three resistant cell lines (TYKNU, WiDR, and A549) were transduced for either control or sgRNAs that targeted BFL-1 or BCL-w. The cells were treated with the BH3 mimetic panel and assessed for viability loss after 3 days in drug. **c** Example of gene ablation. For a representative cell line (A549), knockout of BFL-1 or BCL-w was assessed via western blotting **d** Overlap of outlier phenotype data (BCL-2 gene dependencies) vs outlier mRNA expression. The dependency scores of BCL-2, BCL-X$_L$, and MCL-1 from each outlier cell line were compared to the average scores from the matched tissue type and plotted in a heatmap. Red or blue indicates an increased or decreased dependency score respectively in that cell line compared to the average score for that tissue. For each cell line, the average gene expression data for these genes was compared to the average for that tissue type, and plotted as enriched (red) or depleted (green) relative to tissue-average. **e**, **f** Genetic validation of outlier expression vs outlier phenotype. Two cell lines were selected: H441 cells, which are more dependent on BCL-X$_L$ than other LUAD lines and have above-average NOXA expression, and Panc 03.27 cells, which are more dependent on BCL-X$_L$ than other PAAD lines and have above-average NOXA expression. In both lines, NOXA was depleted and cells were assessed for sensitivity to the BCL-X$_L$ inhibitor (n = 3) and data are presented as mean viability loss ± SEM

demonstrated clinical efficacy in hematological malignancies. However, analogous approaches targeting BCL-2 in solid malignancies have been far less successful, with only a handful of success stories[17,28]. Thus, there is a critical need to understand the contexts in which BH3 mimetics targeting BCL-2, BCL-X$_L$, MCL-1, and other family members have activity, either as single agents or in defined combinations.

Here, addressing this need, we defined the landscape of BCL-2 gene dependencies in cancers derived from ten distinct tissues of origin. This effort identified both known and novel single-gene dependencies as well as combinatorial co-dependencies. Surprisingly, we observed unexpected cases of single gene dependencies across cell lines from a variety of tissues. For example, we identified a novel BCL-X$_L$ dependence in the majority of tested BLCA

cell lines as well as roughly 50% of PAAD cell lines. These findings demonstrate that, although rare, acute sensitivity to single agent BH3 mimetics can occur in solid cancers. Further, we discovered that the expression levels of the BCL-2 family proteins BCL-2, BCL-X$_L$, and NOXA are associated with, and functionally regulate, sensitivity to single agent BCL-2, MCL-1, and BCL-X$_L$ inhibitors across tumors from diverse tissues of origin. These findings confirm and extend previous findings. For example, in LAML, it is appreciated that BCL-2 expression levels predict sensitivity to BCL-2 inhibition, and recent studies in breast cancer imply that BCL-X$_L$ function is critical for sensitivity to MCL-1 inhibition[4–6]. Together, the discovery that solid tumors often harbor exquisite sensitivities to single agent BH3 mimetics targeting MCL-1 or BCL-X$_L$, and the discovery that these dependencies can be accurately predicted based on the expression of the BCL-2 family members BCL-X$_L$ and NOXA, respectively, may have direct translational implications.

Another striking feature of this dependency landscape is the pervasive and synergistic efficacy of BCL-X$_L$ + MCL-1 co-inhibition across cell lines from multiple tissues of origin. Specifically, this combination was synergistically active in roughly 50% of solid tumor cell lines, a discovery that extends the results of recent reports identifying a BCL-X$_L$/MCL-1 co-dependency in breast and small cell lung cancers[4–7]. To better understand the basis for this synergy, we performed GSEA analysis, coupled with functional validation experiments, ultimately identifying a signaling pathway linking EMT to PERK signaling which culminates in increased NOXA expression and functional dependence on BCL-X$_L$. Thus, mesenchymal-like cells have increased BCL-X$_L$ dependence, whereas epithelial-like cells are characterized by synergistic co-dependence on BCL-X$_L$ and MCL-1. It is well established that EMT can promote resistance to a variety of chemotherapies and cellular stresses through the acquisition of stem cell-like properties, including resistance to apoptosis, and recent reports imply that BCL-X$_L$ may drive EMT-induced resistance to apoptosis[19,29]. Our results extend these findings by demonstrating that the process of EMT renders cells dependent on BCL-X$_L$ for their survival. As such, although cells that have undergone an EMT are more resistant to diverse chemotherapies, they exhibit increased sensitivity to BCL-X$_L$ inhibition. Finally, prior studies have established that the PERK-NOXA signaling pathway driving BCL-X$_L$ dependence can be induced by specific chemotherapies, suggesting that these agents may be broadly useful as potentiators of sensitivity to BCL-X$_L$ inhibitors[21,24,30].

Finally, although the growing armamentarium of drugs targeting BCL-2, BCL-X$_L$, and MCL-1 has broadened the landscape of tumors amenable to targeting with BH3 mimetics, our findings demonstrate that some tumors remain insensitive to the combined inhibition of all three of these targets. Using an unbiased, negative-selection CRISPR screen, we identified and validated the BCL-2 family members BCL-w and BFL-1 as potent drivers of intrinsic resistance. As such, these data provide a clear rationale to develop selective, potent inhibitors of BCL-w and BFL-1 in order to further expand the applicability of BH3 mimetic therapies for solid tumors.

## Methods

**Cell culture and reagents**. Refer to Supplementary Table 1 for a full list of all cell lines used in this study and their growth media. All commercially available cell lines were obtained from ATCC, except OVSAHO which was obtained from the JRCB, tested for mycoplasma contamination, and authenticated with the Promega PowerPlex 18D kit for STR profiling. A subset of patient-derived cell lines were either generated at Duke University[12] or at John Hopkins University[31]. All cell lines were grown at optimal confluency (no lower than 30% or higher than 80%) and were given fresh media every 2–3 days. For BH3 mimetics, ABT-199 (obtained from Selleckchem), WEHI-539 (obtained from ApexBio), and

A-1210477 (obtained from Active Biochem) were used as BCL-2, BCL-X$_L$, and MCL-1 inhibitors, respectively. The PERK inhibitor GSK2606414 was purchased from Tocris Bioscience and used at the indicated concentrations.

**BH3 mimetic panel**. Concentrations of ABT-199, WEHI-539, and A-1210477 were carefully selected to yield maximal inhibition (close to 100%) of their annotated targets without significant off-target effects. For ABT-199 and WEHI-539, 1 μM was used, and these concentrations are in line with those used by other groups[26,32]. A-1210477 is less potent than ABT-199 and WEHI-539 and thus a concentration of 10 μM was selected, which is consistent with previous reports[4]. A 500x stock of each of the following combinations of drugs was made at the selected concentrations: ABT-199, WEHI-539, A-1210477, ABT-199 + WEHI-539, ABT-199 + A-1210477, WEHI-539 + A-1210477, ABT-199 + WEHI-539 + A-1210477. All of these combinations yielded a final DMSO concentration of 0.2%, and therefore 0.2% DMSO was used as a vehicle control in the panel. Cell lines were then seeded in triplicate at 5000 cells per well in Greiner white-bottom 96 well plates, and allowed to adhere overnight. The next day, cells were incubated with the BH3 mimetic panel and viability was assessed via Cell Titer Glo (Promega) after a 72 h incubation in drug. This drug incubation period was selected based on previous optimization experiments in our lab. Percentage viability loss was calculated as follows: 100*[1.0 – ((average signal for a given treatment) / (average signal from untreated wells))]. This turned viability loss into a positive signal ranging from 0 to 99 (with 0 indicating no apparent viability loss) and this signal was used in the various heatmap figures and bar graphs and is also referred to in the text as "dependence/dependency" scores. For calculating synergy of the BCL-X$_L$ + MCL-1 combination, we first divided all dependency scores by 100 to convert each value to a 0 to 1 scale, and used the Bliss formula: Synergy = (1- BCL-X$_L$ dependence) * (1-MCL-1 dependence) – (observed BCL-X$_L$ + MCL-1 co-dependence).

*In vivo experiment*. All animal studies were performed at Duke University under an Institutional Animal Care and Use Committee (IACUC) approved protocol, and all studies adhered to the outlined guidelines on ethical usage of research animals. JH4.3 cells were dissociated in 0.25% typsin, washed twice in PBS, and then resuspended in PBS/Matrigel (Corning, PBS: Matrigel = 50: 50%) solution at 2.5 × 10$^7$ cell/ml on ice. 100 μl of cell suspension were injected subcutaneously into the right flank of each 7-week-old male Athymic nu/nu mice (Duke Breeding Core). Tumors were measured twice a week using a Vernier caliper and volumes were calculated using the formula, $V = \frac{(L^2 \times W)}{2}$ ($L$ = longest diameter, $W$ = shortest diameter)). When tumor size reach about 60–100 mm$^3$, the mice were enrolled into A-1331852 treatment group or vehicle group at random. Each group contains 6–7 mice, a number expected to be sufficient to detect statistically significant differences using a one way ANOVA followed by the Student Newman Keuls test ($p < 0.05$, confidence of 0.9, s/delta of 0.3). A-1331852 was formulated in 60% Phosal 50PG, 27.5% PEG400, 10% ethanol, and 2.5% DMSO. Mice were administrated 100 μl of A-1331852 solution (prepared at 25 mg/kg) or vehicle by oral gavage every day. Endpoint of the study was determined using time to reach tumor volume of ~1000 mm$^3$ or tumor ulceration. When tumors reached the endpoint, mice were euthanized under $CO_2$ and tumors were harvested immediately.

**Statistical analysis: tissue vs. mutation status**. To test if oncogene or tumor suppressor status correlated with BCL-2 gene dependency data, cell lines were first binned as either wild-type or mutant (including amplifications/deletions) for the following genes: *TP53, PIK3CA, PTEN, KRAS, EGFR, NF1, BRAF, RB1, ATM, BRCA2,* and *BRCA1*. To investigate whether genetic mutation status or tissue of origin is a better predictor of drug effectiveness, we used the following linear modeling set up:

$$y = X\beta + \varepsilon, \ \varepsilon \sim N(0, \sigma^2 I), \tag{1}$$

where **y** is an *n*-vector of drug sensitivity scores, **X** is an $n \times p$ design matrix jointly holding both cell line tissue of origin and mutation status of select oncogenes, $\beta$ is the corresponding *p*-dimensional vector of additive effect sizes, and $\varepsilon$ is normally distributed residual noise with variance $\sigma^2$ and identity matrix **I**. For each single agent and drug combination, we compute a *p*-value denoting the significance of association for each tissue type and oncogene mutation of interest (See Supplementary Table 2). To further prove the robustness of these results, we also conducted an alternative analysis where we compared how well each biomarker explained the variation across the drug sensitivity scores (see "Goodness of fit", Supplementary Table 2). Here, we used the following to compute an R-squared ($R^2$) statistic, which assessed how well each data type fit model:[1]

$$R^2 = 1 - \frac{SS_{res}}{SS_{tot}}; \ SS_{res} = \sum_i (y_i - x_i^T \beta)^2; \ SS_{tot} = \sum_i (y_i - \bar{y})^2; \tag{2}$$

where $\bar{y}$ is the mean of the observed drug sensitivity scores. Note the greater the $R^2$ statistic, the better the model fit.

**Explaining drug sensitivity with gene expression**. For cell lines included in the CCLE, mRNA expression data ("Gene-centric RMA-normalized mRNA expression data") was obtained from the CCLE download portal (https://portals.broadinstitute.

org/ccle/home). We again used the linear model in Eq. (1) to assess which gene is the best predictor of single agent drug effectiveness. In this case, $X$ is now an $n \times p$ design matrix jointly containing the $\log_2$-transformed expression values of selected biomarker genes. Once again, for each drug, we compute a p-value for each gene denoting the significance of association (see Supplementary Table 3). Next, for each of the most significantly associated genes, we investigated the threshold level of expression needed to reliably predict that a given sample will exhibit dependence to a single agent. Here, we use a sliding window to bin cell lines into groups according to those that express associated genes above and below the specified threshold. We then use a t-test to compute a p-value which describes the degree of significance between the two groups (see Fig. 2b). To independently test if the NOXA expression threshold level of 10.33 ($\text{LOG}_2$ mRNA expression value) also predicts BCL-$X_L$ dependence in a larger data set, we mined NOXA expression and ABT-263 (a dual BCL-2/BCL-$X_L$ inhibitor) sensitivity data from a publicly available dataset[18]. Cell lines were split into two groups, above or below the NOXA expression threshold of 10.33, and ABT-263 IC50 values (LOG2) were plotted in box whisker plots. A t-test was performed to test for significance between the two groups ($p < 0.0001$)

**Gene set enrichment analysis**. To investigate gene set enrichment, we used the global test program[33] to derive multiply corrected p-values of absolute enrichment across a select subset of previously experimentally validated gene signatures on the molecular signature database (MSigDB) version 6.0 compiled at the Broad Institute[34]. These collections included: (A) the H (hallmark) gene sets, and (B) the C2 CP (canonical pathways), BioCarta, KEGG, and Reactome gene sets. To identify significantly enriched gene sets, we first derived a self-curated signature of genes associated with BCL-2/BCL-$X_L$/MCL-1 dependence using the Bayesian approximate kernel regression model[35]. This nonlinear framework identifies differentially expressed genes while simultaneously considering interaction effects between genes, as well as tissue specific effects between samples. In the context of the current study, we assessed drug synergy by regressing the RMA-normalized mRNA gene expression of each cell line[36] onto the GI50 value of each inhibitor combination. Members of the self-curated gene signature were then determined by measuring the magnitude of the treatment effect onto the expression of each gene (posterior probabilities of association (PPA) > 0.5)[37]. These self-curated signature genes were passed through as input in the global test program. A gene set was differentially enriched if it had a Benjamini–Hochberg corrected p-value (BH q-value) below 0.05.

**Genetic manipulation using ORFs, shRNA, and CRISPR**. For overexpression of BCL-$X_L$, we used a previously published V5-tagged BCL-$X_L$ ORF plasmid[38]. An ORF containing HcRED was used as a control. The shRNA and CRISPR sequences used are listed in Supplementary Table 6:

Functional CRISPR/Cas9 constructs were generated via the Gibson assembly as described[6]. Briefly, 5 μg of LCV2 vector was digested with ESP3I restriction enzyme at 37 C° for 2 h, resolved on a 1% agarose gel, and extracted using a Qiagen Gel extraction kit as per manufacturer's protocol. sgRNAs were amplified using the NEB Phusion Hotstart Flex kit and the following PCR conditions: 98 C° for 30 s, [98 C° for 10 s, 63 C° for 10 s, 72 C° for 15 s] x 18 cycles, 72 C° for 3 min.

The following array primers were used:

Array Forward = TAACTTGAAAGTATTTCGATTTCTTGGCTTTATATAT CTTGTGGAAAGGACGAAACACCG

Array Reverse = ACTTTTTCAAGTTGATAACGGACTAGCCTTATTTTAA CTTGCTATTTCTAGCTCTAAAAC

Amplified inserts were then cleaned using the Axygen AxyPrep Mag PCR Clean-up kit as per manufacturer's protocol. 100 ng of cut LCV2 vector were combined with 40 ng of each sgRNA and ligated using the Gibson Assembly protocol as per manufacturer's protocol. The assembled sgRNA constructs were electroporated into e cloni bacteria, and spread over LB plates containing ampicillin (amp) overnight. Individual colonies were selected from these plates, inoculated in LB amp for 16 h, and the plasmid DNA was prepped using the Qiagen plasmid mini-prep protocol.

Virus was produced using the same protocol for the ORF, shRNA, and CRISPR constructs. Briefly, 2.75 μg of plasmid construct DNA was mixed with the packaging plasmids PSPAX2 (2.75 μg) and VSVG (0.275 μg) in 17 μL of FuGene 6 and Opti-MEM (Gibco) up to a final volume of 280 μL. This mixture was incubated at room temperature for 30 min and then added drop-wise to a 10 cm dish containing 293 T cells grown to ~30% confluency. After 18 h, the media was replaced with 20 mL of harvest media (DMEM (Gibco) + 1% penicillin/streptomycin + 30% fetal bovine serum) and the harvest media was collected after 48 additional h and filtered (0.45 μM) via syringe. For transductions of ORF and shRNA virus, $0.1 \times E6^6$ cells were seeded in a 6 well plate and were allowed to attach for 24 h. Cells were then transduced using 0.2–0.5 mL of virus + RPMI 1640 (with 1% pen/strep and 10% FBS) to a final volume of 2 mL. Polybrene was added to a final concentration of 16 μg/mL. The plates were then centrifuged (1,126 RCF, 37 C°) for 1 h. Afterwards, the virus-containing media was replaced 2 mL of the cell line specific media. The next day, the media was replaced with fresh media containing 2 μg/mL puromycinmycin (puromycin) and the plates were incubated an additional 48 h in puromycin to kill non-transduced cells. A non-transduced well of cells was included to ensure adequate kill of non-transduced cells. For cells transduced was CRISPR/Cas9

constructs, the same procedure was done except that after 2 days in puromycin, cells were moved to 10 cm or 15 cm dishes and allowed to grow in puromycin for an additional 5 days. This ensures adequate time for CRISPR-mediated genetic ablation. For the experiments testing the effects of the BCL-$X_L$ ORF, NOXA shRNA, E-cadherin shRNA, BFL-1 CRISPR, BCL-w CRISPR; cells were plated in 96 well plates and treated with the BH3 mimetic panel as described in the above method section. Protein lysates were generated from each condition to provide examples of each genetic manipulation.

**Protein lysates, western blotting, and densitometry**. Whole-cell protein lysate generation, protein quantitation via Bradford assay, and western blotting was performed as described[6]. Briefly, cell pellets were resuspended in lysis buffer, rotated at 4 C° for 10 min, and cleared of debris via 16,200 RCF centrifugation for 10 min at 4 C°. Lysates were then quantitated using a standard Bradford assay and denatured using 3x NuPage buffer. Westerns blots were run on a NuPage gradient gel. The following antibodies were used at 1:1000 dilutions, except NOXA and ATF3 which were used at 1:500: Vinculin (Cell Signaling Technologies #4650), BCL-$X_L$ (Cell Signaling Technologies #2764), NOXA (Santa Cruz Biotechnology #114C307), E-Cadherin (Cell Signaling Technologies #3195), BCL-$X_L$ (Cell Signaling Technologies #2764), PERK (Cell Signaling Technologies #5683), p-eIF2α (Cell Signaling Technologies #3398), ATF4 (Cell Signaling Technologies #11815), ATF3 (Santa Cruz Biotechnology sc-188), β-actin (Cell Signaling Technologies #4970), BCL-w (Cell Signaling Technologies #2724), BFL-1 (Cell Signaling Technologies #14093), SLUG (Cell Signaling Technologies #9585). Rabbit or mouse HRP-conjugated secondary antibodies (#7074, #7076) were used at a dilution of 1:2000 and proteins were detected using standard ECL detection. To quantify relative expression of SLUG protein, we first used densitometry to quantify SLUG and actin levels from each tested cell line. Normalized SLUG expression was then compared amongst all cells lines and these data were used for the Pearson Correlation analysis.

**BH3 profiling**. The BH3 profiling assay was performed as previously described[39]. In brief, cells were digitonin permeabilized and incubated with fluorescent mitochondrial dye (JC-1) and fluorescence was analyzed over time in a 384 well plate format. Mitochondrial priming was measured based on changes in depolarization over time. Peptides from the BH3 domains of BIM, PUMA, BAD, NOXA, and HRK were used to determine the priming states of the analyzed cell lines.

**Epithelial and mesenchymal gene signatures**. For cell lines with mRNA expression data (CCLE cell lines), we quantified an epithelial and mesenchymal gene score for each cell line by summing the expression levels of either selected epithelial or mesenchymal markers. For epithelial markers, we used: MUC1, CDH1 (E-cadherin), EPCAM, and CLDN3. For the mesenchymal markers, we used: SNAIL, SLUG, TWIST1, TWIST2, VIM, and CDH2 (N-cadherin). These genes are all established, canonical markers of epithelial vs. mesenchymal states. These signatures were also used to calculate average signature scores for each tissue type used in this study.

**Negative-selection pooled CRISPR screen**. Pooled screening was performed as described[40]. Briefly, $2 \times 10^6$ TYKNU cells were transduced with a pooled library of barcoded CRISPRs that target 398 genes important in cancer (5x sgRNA per gene plus 50 control sgRNAs). A multiplicity of infection (MOI) of 0.3 was used to reduce the chances of cells taking up more than one CRISPR construct. After selection in puromycin for ~10 days, cells were plated at $>1 \times 10^6$ cells per condition (this ensures 1000x coverage of each CRISPR construct). The conditions were performed in duplicate and were either 0.2% DMSO or 1 μM WEHI-539+5 μM A-1210477. After 2 weeks of treatment, DNA was harvested from each condition and cleaned via the Axygen AxyPrep Mag PCR Clean-up kit. Two rounds of nested PCR were performed to amplify each CRISPR and to attach sample-unique barcodes. Ninety ng of DNA from each barcoded sample were pooled and sent for deep-sequencing to deconvolute the individual counts for each gene from each condition. The depletion metric (DM) for each gene was calculated based on the relative levels of the best 3 sgRNA (three-score) from each gene in the treated sample relative to the DMSO control sample. Relevant genes involved in apoptosis were highlighted as follows: red, BCL-w (BCL2L2) and green, BFL-1 (BCL2A1).

## Data availability

All data are available upon request from the corresponding or lead authors.

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

## Acknowledgements

This work was supported by a Liz Tilberis Early Career Award from the Ovarian Cancer Research Fund Alliance, a Department of Defense Breast Cancer Research Program Breakthrough Award (W81XWH-16-1-0703), and NIH award R01CA207083 (all to K.C.W.). This work was also supported by an NIH F32 Postdoctoral Fellowship Award (F32CA206234 award to R.S.S.). Any opinions, findings, and conclusions or recommendations expressed in this material are those of the authors(s) and do not necessarily reflect the views of the NIH or DOD.

## Author contributions

Conceptualization: R.S.S and K.C.W.; data generation: R.S.S., E.L., M.L., A.A., G.R.A.; statistical analysis: L.C.; animal studies: M.L.; formal data analysis: K.H.L., M.C.; CRISPR library generation: P.S.W.; writing of original draft: R.S.S., and K.C.W.; writing, review, and editing: all authors; funding acquisition: K.C.W. and R.S.S.; supervision: K.C.W.

## Additional information

**Competing interests:** The authors declare no competing interests.

