## [Peer Review File · Nature Communications]

Reviewers' comments:

Reviewer #1 (Expertise: BCL-2 inhibition, apoptosis, cancer, Remarks to the Author):

Soderquist et al. report an analysis of 91 cancer cell lines, including 13 primary cultures from PDX models or tumors, for cytotoxicity with selective BCL-2, BCL-XL and MCL-1 inhibitors as single agents and in combination. Their analysis indicates an association of epithelial-type cell lines with sensitivity to combined BCL-XL and MCL-1 inhibitors, mesenchymal-type cell lines with sensitivity to the single agent BCL-XL inhibitor, and hematopoietic malignancies with BCL-2 inhibitor sensitivity. They also find that EMT triggers PERK-dependent expression of the MCL-1-selective BH3-only protein NOXA, and that BFL-1 and BCL-w expression may account for resistance to triple inhibitor therapy.

1. Given the known discordance in response to ABT-737 between established SCLC cell lines (in vivo/in vitro) and primary SCLC xenografts (Hann CL, Cancer Res 2008), the authors should confirm in vivo single agent sensitivity in one of the PDX models (e.g. JH 2.5 or TSO III).
2. The authors report GSEA analysis to investigate the basis for BCL-XL and MCL-1 inhibitor synergy "which could not be separated from single agent activity on the basis of BCL-2 family expression correlates". Yet, the EMT signature identified is associated with changes in NOXA expression at the protein, and presumably, RNA level, based on the publications cited. Thus, BCL-2 family expression appears to correlate with the synergy phenotype.

Reviewer #2 (Expertise: Genetics, epigenetics, drug sensitivity, Remarks to the Author):

Soderquist and colleagues present a small molecule assay for single and combination sensitivity to BCL2 family inhibitors across many cancer cells. They report gene expression and cell state correlates of response, in some cases related to EMT programs. They also performed a targeted CRISPR screen for mediators of drug response. This is a clinically important study, as numerous BCL2 family targeting agents are either already in clinical use or in trials across many cancer types. The field needs additional information to guide selection of cancers for treatment, and to better understand why certain tumors do or do not respond. This could be a valuable study to stimulate basic research and improve design and correlative studies on clinical trials. I would ask the authors to address the following comments:

1. Almost all data in the paper is using one small molecule targeting each of, BCL2, BCL-XL, and MCL1. Thus, is it accurate to say the findings are BCL2/BCL-XL/MCL1 dependencies, or rather more appropriately venetoclax/WEHI-539/A-1210477 dependencies? One way to broaden the conclusions would be to test additional small molecules putatively targeting these proteins, and/or adding genetic confirmation with shRNA/CRISPR or similar approach. E.g., do the drug combination findings hold up for two targets with reciprocal compound + CRISPR and CRISPR + compound and CRISPR + CRISPR. Because of this concern, I would be careful with the term "single gene" dependencies as currently throughout the manuscript, rather these are "single compound" dependencies. As such, the title is not technically correct to say "BCL-2 gene dependencies."
2. In situations where all 3 compounds, even in combination, failed to kill cells, the authors hypothesized two possibilities: decreased overall priming or alternative BCL2 family members being important. They seem to jump to the possibility of alternative genes to justify the CRISPR screen. Was overall priming tested in all cell lines?

3. The methods for the (lack of) association experiment between mutation and drug sensitivity are not clear. Was this on a mutation by mutation basis, each tested for association with each compound sensitivity? Was there adequate power to say with statistical significance that there is a lack of association, rather than what I think was actually found: no statistical evidence of an association. I don't think this point is critical to their conclusion, but the methods and language could be more precise. Along these lines, Figure 2B and Supplemental Table 2B seem to be identical – are there supposed to be additional data in the table that could clarify the questions above about methods and power?

4. I don't understand what is plotted in Figure 3D. The x-axis data seems like it should be the same in the left and right panels based on methods and legend. Why are some data points different in their BCL-XL:MCL-1 synergy score between the two panels (e.g., BLCA near zero synergy score on left but 0.3 on right; GBM near zero on left but 0.4 on right)?

5. CRIPSR screen: I did not see a list of the 398 genes and controls tested, nor the primary results of the screen on a gene by gene or sgRNA by sgRNA basis in a supplementary table. These data would be helpful to interpret the results, including to validate the controls, and to understand which sgRNAs conferred resistance (i.e. BCL-XL and MCL-1 would be expected). The full results of the screen should be included with the publication, even if the focus of follow up experiments here is only on BFL-1 and BCL-w.

Reviewer #1:

We thank reviewer #1 for his or her thoughtful comments on this work. Below, we address each point raised in this review.

Comment 1: "Given the known discordance in response to ABT-737 between established SCLC cell lines (in vivo/in vitro) and primary SCLC xenografts (Hann CL, Cancer Res 2008), the authors should confirm in vivo single agent sensitivity in one of the PDX models (e.g. JH 2.5 or TSO III)."

Response 1: The reviewer raises an excellent point, that drug sensitivity observed in immortalized cell lines *in vitro* may not always equate to *in vivo* efficacy. Moreover, as was the case with ABT-737 in SCLC, drug sensitivities observed in established cell lines are not always replicated in primary cell lines or PDX mouse models. In the original submission, we attempted to preemptively address this issue by using several patient derived cell lines from pancreatic (PAAD) and colorectal (COAD) cancers and comparing their results to established cell lines (see Fig. 1D). Encouragingly, the distribution of phenotypes observed in the panel of primary lines mirrored what was observed in the established cell lines (for example, ~50% of PAAD lines responded well to single agent BCL-X_L inhibition; Fig. 1C-D). In response to the reviewer's suggestion, we utilized one of these primary cell lines – the PAAD line JH4.3 – to create an *in vivo* model (Fig. 1E). Consistent with our *in vitro* data showing intermediate single agent sensitivity to BCL-X_L inhibition, we also observed moderate *in vivo* sensitivity to the BCL-X_L inhibitor A-1331852 in this model. (Parenthetically, we note that we first attempted to create xenografts from two of the more sensitive PAAD cultures, JH 2.5 and TSO III, but were unable to form tumors that grew with reliable kinetics.) We thank the reviewer for this suggestion, as the data further validate the findings of this study and the potential of BCL-X_L as a therapeutic target in PAAD.

Comment 2: "The authors report GSEA analysis to investigate the basis for BCL-X_L and MCL-1 inhibitor synergy "which could not be separated from single agent activity on the basis of BCL-2 family expression correlates". Yet, the EMT signature identified is associated with changes in NOXA expression at the protein, and presumably, RNA level, based on the publications cited. Thus, BCL-2 family expression appears to correlate with the synergy phenotype"

Response 2: We apologize for the confusion in the wording mentioned here. To clarify, the GSEA was performed in an effort to uncover signaling network(s) which might explain the BCL-X_L + MCL-1 synergy in an unbiased manner. In this way, we hoped to identify novel drivers of synergy, and then determine if they functioned through regulation of BCL-2 family members. Through this approach, we nominated EMT as a dominate driver of synergy, and then later characterized NOXA as the critical effector protein responsible for EMT-driven changes in synergy. To clarify this in the text, we now state the following: "To identify signaling events that may underlie this BCL-X_L/MCL-1 synergy, gene set enrichment analysis (GSEA) was performed (Fig. 3B)"

Reviewer # 2:

We thank reviewer #2 for his or her thoughtful comments. Below, we address each point raised in this review.

Comment 1: “Almost all data in the paper is using one small molecule targeting each of, BCL2, BCL-XL, and MCL1. Thus, is it accurate to say the findings are BCL2/BCL-XL/MCL1 dependencies, or rather more appropriately venetoclax/WEHI-539/A-1210477 dependencies? One way to broaden the conclusions would be to test additional small molecules putatively targeting these proteins, and/or adding genetic confirmation with shRNA/CRISPR or similar approach. E.g., do the drug combination findings hold up for two targets with reciprocal compound + CRISPR and CRISPR + compound and CRISPR + CRISPR. Because of this concern, I would be careful with the term “single gene” dependencies as currently throughout the manuscript, rather these are “single compound” dependencies. As such, the title is not technically correct to say “BCL-2 gene dependencies.””

Response 1: The reviewer raises an important question – can one equate sensitivity to a selective and potent drug to a dependency on that drug’s target? For example, does ABT-199 sensitivity equate to BCL-2 gene dependency? In this manuscript, we used three different BH3 mimetics as molecular probes to test for “dependence” on their respective targets, and we were careful to use each agent at concentrations that yield near-complete inhibition of their targets while avoiding off-target effects on other BCL-2 family proteins. However, the reviewer is absolutely correct that we cannot know whether these drugs may have other off-target effects that impact our ability to ascribe their activities to their known targets. Following the reviewer’s advice, we addressed this issue by testing three structurally distinct BH3 mimetics (the BCL-2/BCL-X_L dual inhibitor ABT-737, the BCL-X_L inhibitor A-1331852, and the MCL-1 inhibitor S63845) in cell lines we previously deemed as BCL-2, BCL-X_L, or MCL-1 dependent, as well as in a cell line resistant to all combinations of mimetics (see Fig. S2). Reassuringly, the data obtained with these additional BH3 mimetics phenocopied our previous data obtained with ABT-199, WEHI-539, and A-1210477. Thus, we believe we can confidently assert that sensitivity to the BH3 mimetics studied in this work - ABT-199, WEHI-539, and A-1210477 – reflect true dependencies on the targets of these drugs, BCL-2, BCL-X_L, and MCL-1, respectively.

Comment 2: “In situations where all 3 compounds, even in combination, failed to kill cells, the authors hypothesized two possibilities: decreased overall priming or alternative BCL2 family members being important. They seem to jump to the possibility of alternative genes to justify the CRISPR screen. Was overall priming tested in all cell lines?”

Response 2: The reviewer raises an important point – that the failure of certain cell lines to undergo apoptosis in the presence of combined BCL-2/BCL-X_L/MCL-1 inhibition may be caused either by alternative BCL-2 family members supporting survival or, alternatively, a lack of overall priming. To address this, we selected several cell lines that were either sensitive to single agent BH3 mimetics, exhibited BCL-X_L + MCL-1 synergistic dependence, or were resistant to all BH3 mimetic combinations (resistant), then performed BH3 profiling with the BIM and PUMA peptides to measure overall priming levels (see Fig. S3). Importantly, we found that all three groups exhibited BIM-induced depolarization, indicating that even cells in the resistant group were capable of undergoing BAX/BAK-mediated depolarization. The single agent sensitivity group exhibited a statistically higher level of PUMA-induced depolarization than the synergistic group, indicative of a higher level of priming in this group of cell lines. However, the resistant cell line group exhibited comparable levels of overall BCL-2 priming (via the PUMA signal) as the synergistic group. Together, these data indicate that the resistant cells possess some degree of dependence on BCL-2 family anti-apoptotic proteins and are apoptosis-competent. Further supporting this notion is the fact that apoptosis has been reported in each of these resistant cell lines in the published literature (cited in the manuscript) in response to a variety of chemotherapeutic agents. Based on these findings, we conclude that resistant cell lines likely rely on alternative BCL-2 genes (e.g., BCL-w, BFL-1) for survival. Ultimately, this notion was

supported by data from the CRISPR/Cas9 screens and associated follow up experiments, which confirmed that knockout of BCL-w and BFL-1 sensitizes the group of resistant cells to BH3 mimetics.

Comment 3: “The methods for the (lack of) association experiment between mutation and drug sensitivity are not clear. Was this on a mutation by mutation basis, each tested for association with each compound sensitivity? Was there adequate power to say with statistical significance that there is a lack of association, rather than what I think was actually found: no statistical evidence of an association. I don’t think this point is critical to their conclusion, but the methods and language could be more precise. Along these lines, Figure 2B and Supplemental Table 2B seem to be identical – are there supposed to be additional data in the table that could clarify the questions above about methods and power?”

Response 3: We apologize for these points of confusion. As correctly noted by the reviewer, Supplemental Table 2B did indeed contain the same data that was used in Fig. 2B. This redundancy was intentional but we see now how it can be confusing and as such have removed it from the Supplemental Table. In addition, we would like to clarify our statistical analysis of BCL-2 gene dependency correlations with oncogene mutation status vs. tissue of origin. Our main point was to determine the best predictor (mutation status vs. tissue of origin) for each dependency state (e.g. BCL-2, vs BCL-X_L, etc.). The data shown in Figure 2B show the R² values for each phenotypic class overlaid with either mutation status or tissue of origin. In all cases, tissue of origin was the dominant predictor. However, as correctly stated by the reviewer, we are not stating that mutation status never correlates with BCL-2 gene dependencies; just that tissue of origin appears to be a stronger predictor within this dataset. We have adjusted the text for this section to clarify this issue by stating, “Two of the most dominant drivers of phenotype in a given cancer are oncogenic mutations (oncogene or tumor suppressor status) and tissue of origin. As such, we first sought to determine which of these two properties best predicted BCL-2 gene dependencies.”

Comment 4: “I don’t understand what is plotted in Figure 3D. The x-axis data seems like it should be the same in the left and right panels based on methods and legend. Why are some data points different in their BCL-X_L:MCL-1 synergy score between the two panels (e.g., BLCA near zero synergy score on left but 0.3 on right; GBM near zero on left but 0.4 on right)?”

Response 4: We apologize for this issue and thank the reviewer for pointing out this error. The labels had been mislabeled in the second graph in Figure 3D and have now been corrected. The interpretation of this plot in the manuscript text remains correct. Further, we have carefully double-checked the other figure axes throughout the manuscript and supplemental information to ensure that no similar oversights occurred.

Comment 5: “CRISPR screen: I did not see a list of the 398 genes and controls tested, nor the primary results of the screen on a gene by gene or sgRNA by sgRNA basis in a supplementary table. These data would be helpful to interpret the results, including to validate the controls, and to understand which sgRNAs conferred resistance (i.e. BCL-X_L and MCL-1 would be expected). The full results of the screen should be included with the publication, even if the focus of follow up experiments here is only on BFL-1 and BCL-w.”

Response 5: We thank the reviewer for this point and have now included the results from our screen in Supplemental Table 4. Specifically, we have included the full sgRNA library list, the 3-score gene depletion mean values for each replicate, the average depletion scores, and the

ranked list of genes from most to least depleted based on depletion average. We have highlighted the genes of interest, BCL-w and BFL-1, in red.

REVIEWERS' COMMENTS:

Reviewer #1 (Remarks to the Author):

I am satisfied with the responses to my critiques.

Reviewer #2 (Remarks to the Author):

The authors have addressed all of my concerns, and I commend them on this interesting work. These data will make an important contribution to the field's efforts to optimize use of BH3 mimetics for cancer therapy.